# Effect of Deglycosylated Rutin by Acid Hydrolysis on Obesity and Hyperlipidemia in High-Fat Diet-Induced Obese Mice

**DOI:** 10.3390/nu12051539

**Published:** 2020-05-25

**Authors:** Jinwoo Yang, Junsoo Lee, Younghwa Kim

**Affiliations:** 1Division of Food and Animal Sciences, Chungbuk National University, Cheongju, Chungbuk 28644, Korea; wlsdnwow@naver.com (J.Y.); junsoo@chungbuk.ac.kr (J.L.); 2School of Food Biotechnology and Nutrition, Kyungsung University, Busan 48434, Korea

**Keywords:** rutin, aglycone, quercetin, deglycosylation, obesity

## Abstract

The present study evaluated the effects of acid-treated rutin on hyperlipidemia and obesity in high-fat diet (HFD)-induced obese mice. The mice consumed a HFD with or without acid-treated rutin for 7 weeks. Body weight gain considerably decreased, by approximately 33%, in the acid-treated rutin (AR) and quercetin (Q) groups compared to that in the HFD group. The adipocytes’ size in epididymal fat in AR and Q groups was significantly reduced compared to that in the HFD group (*p* < 0.05). Treatment with AR decreased the levels of triglycerides, total cholesterol, and low-density lipoprotein cholesterol compared to the HFD group. In particular, administration of AR significantly decreased serum triglyceride (36.82 mg/dL) by 46% compared to HFD (69.30 mg/dL). The AR group also showed significantly decreased atherogenic indices and cardiac risk factors. These results suggest that deglycosylated rutin generated by acid treatment enhances the anti-obesity and hypolipidemic effects in obese mice, and provides valuable information for improving the functional properties of glycosidic flavonoids.

## 1. Introduction

Metabolic syndrome is related to blood lipid disorders, diabetes mellitus, obesity, and cardiovascular disease [1]. In particular, obesity is a crucial health problem caused by excessive caloric intake and low energy expenditure, and the incidence of overweight and obesity has been increasing over the past 50 years [2]. Imbalance between energy expenditure and consumption induces excess lipid accumulation in various organs and tissues, including liver and adipose tissue [3]. Several studies have shown that increased oxidative stress leads to the accumulation of fat in the body, which consequently induces obesity [4,5]. Recently, it was reported that natural bioactive food components with less pronounced side effects could be useful for the control of fat accumulation in liver, hyperlipidemia, and blood glucose concentrations [6,7]. 

Rutin, also known as rutoside, is a flavonol glycoside combining one molecule of quercetin and rutinose each. Rutin is well known to possess various beneficial health effects, including antitumor, anti-inflammatory, anti-obesity, and antioxidant activity [8,9,10,11]. Most natural flavonoids are *O*-glycoside or *C*-glycoside forms found in plants. Previous studies have shown that glycosylated flavonoids show less immune-regulatory activity compared to their aglycone forms because of reduced cellular absorption [12]. Moreover, aglycone flavonoids can easily be absorbed by the small intestine, whereas glycosidic flavonoids must be converted into the aglycone form to be similarly absorbed [13]. Therefore, deglycosylation of flavonoid glycosides would be an important step for increasing their biological activities. A previous study reported by our group showed that acid treatment is effective for the conversion of rutin to quercetin [14]. Quercetin is well known for its anti-obesity property by attenuating adipogenesis through the up-regulation of the adenosine monophosphate-activated protein kinase pathway [15]. In an animal model, quercetin inhibited adipogenesis by decreasing the adipogenic factor expression such as CCAAT/enhancer binding protein α and suppressed lipogenesis by reducing the expression of fatty acid synthase and the activity of acetyl conenzyme A carboxylase [16]. Furthermore, deglycosylated rutin led to enhancement of cytoprotective activity against oxidative stress, anti-inflammatory activity, and anti-adipogenic activity in model cellular systems in our previous study [14]. Therefore, this current study aimed to compare the potential anti-obesity and hypolipidemic effects of rutin and acid-treated rutin in obese mice.

## 2. Materials and Methods 

### 2.1. Chemicals

Rutin (≥94%), quercetin (≥95%), ethyl acetate, hydrogen chloride, and dimethyl sulfoxide (DMSO) were purchased from Sigma-Aldrich, Co. (St. Louis, MO, USA). All other chemicals used were of analytical grade.

### 2.2. Sample Preparation

To prepare acid-treated rutin, a previously reported method was used with slight modifications [14]. Rutin (500 mg) was dissolved in 500 mL of acidic solvent (80% ethanol with 0.5 M HCl). The solution was agitated in a water bath shaker at 75°C for 2 h with a reflux condenser. When the reaction was complete, the reactant was cooled and adjusted to pH 7 using sodium hydroxide (0.1 M) in 80% ethanol. The organic solvent in the neutralized reactant was eliminated by vacuum evaporation. Then, the residue was eluted in distilled water and the aqueous solution was rinsed with ethyl acetate several times until a clear ethyl acetate layer was formed on top. The collected ethyl acetate layer was concentrated under reduced pressure. The residue was dissolved in DMSO and stored at −20 °C until use.

### 2.3. Animal Treatment

Four-week-old male mice (C57BL/6J) were obtained from Daehan Biolink (Chungbuk, Korea) and maintained under standard light (12 h light/dark cycle), relative humidity (55 ± 5%), and temperature (23 ± 1 °C) conditions. After 1 week of acclimatization, the animals were separately fed a normal diet (ND) or high-fat diet (HFD). Table 1 shows the ingredients and composition of the normal and HFDs (Research Diets, Inc., New Brunswick, NJ, USA). Then, the animals were randomly subdivided into five groups (*n* = 7 in each group): ND, HFD, HFD with 100 mg/kg rutin/day (R), HFD with 100 mg/kg acid-treated rutin/day (AR), and HFD with 100 mg/kg quercetin/day (Q) [17]. Quercetin was used as the positive control. Each stock solution of sample was prepared with DMSO, and the stock solution was diluted with 0.5% carboxymethyl cellulose (CMC) water. The final DMSO concentration never exceeded 0.1% (v/v) in any group. Each dietary preparation was orally administered daily throughout the experimental period (7 weeks). Mice in ND and HFD groups were orally administered with vehicle alone (0.5% CMC water containing 0.1% DMSO). Their body weight and food consumption were recorded once a week. The animal experiment was approved by the Animal Care and Use Committee of Chungbuk National University (approved number: CBNUA-1145-18-01), and all experimental procedures were performed in accordance with the guidelines.

### 2.4. Sample Collection

All mice were fasted for 12 h and exposed to diethyl ether after administering samples for 7 weeks. Then, blood for serum biochemistry analysis were collected in blood collection tube (Vacutainer SST tubes, BD Bioscience, Franklin Lakes, NJ, USA). The plasma was obtained by centrifuging the blood at 10,000 × *g* for 5 min at 4 °C. The epididymal adipose tissue and liver were carefully harvested, washed in cold normal saline solution, and weighed.

### 2.5. Histopathology

Liver and epididymal adipose tissue were fixed in 10% phosphate-buffered formalin and processed for paraffin section. The sections (4 μm) were stained with hematoxylin and eosin (H&E) and observed under an optical microscopic system (Axiovert 100, Carl Zeiss, Goettingen, Germany). In addition, H&E-stained epididymal adipose tissues were analyzed to measure fat droplet size using AxioVision Rel 4.8 software (Carl Zeiss).

### 2.6. Measurement of Plasma Biochemical Parameters

Blood urea nitrogen (BUN), creatinine (CREA), alanine transaminase (ALT), aspartate transaminase (AST), glucose (GLC), triglyceride (TG), low density lipoprotein cholesterol (LDL-c), total cholesterol (TC), and high density lipoprotein cholesterol (HDL-c) levels in plasma were measured using a biochemical automatic analyzer (Hitachi 7080, Hitachi Instrument Ltd., Tokyo, Japan).

### 2.7. Statistical Analysis

Data were expressed as means ± standard deviation. Two-way repeated analysis of variance (treatment × time as repeated measures) was conducted using Proc Mixed procedure followed by Tukey–Kramer post-hoc test for mean separations in SAS version 9.4 (SAS Institute Inc., Cary, NC, USA) to analyze the body weight change shown in Table 2. Data, except for body weight change, were analyzed by one-way ANOVA, followed by Duncan’s multiple comparison test. *p*-values less than 0.05 were considered statistically significant. All statistical analyses were performed using a SAS version 9.4.

## 3. Results and Discussion

### 3.1. Effects of Acid-Treated Rutin on Body Weight Gain and Food Intake

In order to examine whether acid-treated rutin affects body weight, five groups of mice were fed with or without samples for 7 weeks. Table 2 shows changes in body weight. Body weight appeared to increase by week. Significant increase in body weight occurred at around 2–3 weeks for all treatments except for the ND, which showed slower increase. There were no significant differences among treatments until 2 weeks. Body weight of HFD and ND groups began to differentiate significantly at week 3 and remained the highest and lowest at week 7, respectively. Body weights of R, AR, and Q groups did not differ at week 7, but were significantly lower than the HFD group and higher than the ND group. The results of food consumption, weight gain, and food efficiency ratio (FER) are shown in Table 3. After 7 weeks, the mice fed an HFD exhibited an approximately threefold increase in body weight gain compared to the ND group. All diets significantly diminished body weight gains compared to HFD-fed animals. During the experiment, food intake was higher in the ND group than the HDF groups, but no significant difference was observed between the HFD group and treatment groups. The FER of the HFD group was significantly higher than that of the ND group. However, there was no statistically significant difference between treatment groups and the HFD group. These results suggested that the reduction in body weight by diet type was not associated with food intake in mice. Obesity leads to adipose tissue dysfunction, which is implicated in increased vascular contractility, inflammation, and endothelial insulin resistance [18,19]. Previously, it was reported that being overweight and obese increases the chances of developing diabetes, as well as heart and liver diseases [20,21]. Therefore, the key to preventing or treating obesity and its associated disorders is managing body and fat weight. The anti-obesity effects of rutin, including weight loss, decreased adipose tissue weight, and reduced blood lipid concentrations have been reported by Hsu et al. [22]. Another study showed that quercetin significantly decreased body weight, perirenal fat, and epididymis fat compared with HFD, but rutin showed no effect on this basis [23]. The inconsistent effect of rutin may be explained by the different dose, background diet, animal model, and duration of supplementation. Most of the rutin was converted to quercetin by acid treatment in previous reports from our group [14]. In the present study, all treatments, including rutin, acid-treated rutin, and quercetin, did not affect food intake, but they significantly decreased body weight gains and, as a result, had potential anti-obesity activities in HFD-induced mouse models.

### 3.2. Effect of Acid-Treated Rutin on Histological Change in Epididymal Adipose Tissue and Liver

Table 4 shows the epididymal adipose tissue and liver weights after the experimental period. The mice exhibited significantly increased liver weight in the HFD group compared to those in the ND group. The weight of epididymal fat tissue in the HFD group was markedly increased compared to that in the ND group. However, epididymal fat weights in treatment groups were significantly decreased compared to those in the HFD group. The weights of liver were significantly lower in all treatment groups than the values for the mice fed HFD only. Moreover, treatment with AR presented animals with the lowest liver weight. The histological changes in liver and epididymal adipocytes were examined by H&E staining. The HFD group showed increased hepatic fat accumulation compared to the ND group (Figure 1A). However, lipid accumulation in liver was reduced in the treatment groups. H&E-stained epididymal fat tissue is shown in Figure 1B. Epididymal adipocytes in the HFD group were markedly enlarged compared to those in the ND group. However, all samples reduced adipocyte sizes compared to no treatment (HFD group). Additionally, relative epididymal adipocyte sizes were analyzed using image analysis software. As revealed by histological analysis, the greatest epididymal adipocyte sizes were observed in the HFD group (Figure 2). Adipocytes sizes in the Q and AR groups were markedly reduced compared to those in the HFD group by approximately 30%, whereas there were no significant differences in adipocytes sizes between the HFD and R groups. Obesity is characterized by increased white fat deposits due to adipocyte hyperplasia and hypertrophy. A previous study showed that epididymal fat tissue is closely related to body weight and body weight gain [24]. Additionally, several natural compounds can inhibit fat accumulation. For example, the weight of white adipose tissue induced by an HFD was decreased by dietary procyanidins through a combination of the agonist-like action of peroxisome proliferator-activated receptor (PPAR)α and antagonist-like action of PPARγ [25]. A previous report showed that quercetin supplementation reduced intrahepatic lipid accumulation significantly through its ability to modulate lipid metabolism gene expressions [26]. In this study, increased fat accumulation and enlarged adipocytes were observed in the HFD group. The administration of AR and Q efficiently suppressed lipid accumulation in epididymal adipocytes and liver. These results are also consistent with previous studies that have addressed quercetin’s action in the liver of mice fed an HFD [26,27]. Therefore, it was considered that conversion of rutin to quercetin by acid hydrolysis resulted in reduced fat accumulation and adipocyte sizes in vivo.

### 3.3. Effect of Acid-Treated Rutin on Serum Lipid Profiles

Biochemical parameters of blood plasma are shown in Table 5. AST and ALT levels reflect the degree of damage to hepatocytes; HFD increased the serum levels of AST and ALT due to accumulation of lipids. The ALT levels were decreased in treatment groups compared with the HFD group. The AST levels were also reduced in treatment groups except for the AR group. Although there was a trend for AST to decrease in the AR group, this effect, however, was not statistically significant. Plasma level of BUN represents kidney function and toxicity, and there were no significant differences across all experimental groups. High levels of CREA in the bloodstream could be an indicator of kidney disease. A previous study showed that an HFD increased the CREA level [23]. In the results, there was no difference between ND and HFD groups in terms of CREA levels, and the administration of R, AR, and Q decreased the CREA levels compared with the HFD. These results indicated that R, AR, and Q did not exert renal toxicity in our murine model system. Generally, HFD-fed mice exhibited significantly increased TC and TG levels in serum [28]. As a result, the administration of AR significantly reduced lipid metabolic parameters such as TG, TC, HDL-c, and LDL-c. In particular, HFD-induced increases in TG and LDL-c levels were dramatically reduced in the AR and Q groups. The TC level in the Q group was decreased compared with HFD and R groups, however, there was no significant difference. In a previous study, it was reported that suppression of TG absorption has a prominent role in weight loss [29]. In addition, a previous report has indicated that quercetin significantly decreased TG and LDL-c; however, rutin did not affect the levels of lipid metabolism parameters [23]. In this study, the administration of AR or Q was effective in lowering TG and LDL-c in vivo, and these results are also consistent with the previous study. Therefore, these results suggest that deglycosylation of rutin by acid treatment could be effective against hyperlipidemia in obese mice, and that the effects of acid-treated rutin are comparable to the effects of quercetin on hyperlipidemia. The risk of cardiovascular disease is usually associated with atherogenic index (AI) and cardiac risk factors (CRFs) [30]. Here, AI and CRF were also calculated (Figure 3). AI and CRF levels in the HFD group were significantly elevated compared to the ND group, however, there were significant decreases in the levels of AI and CRF in the Q and AR groups. In this study, it was founded that AR exerted anti-obesity effects by decreasing weight gain, epididymal adipocyte weight and size, and levels of AI and CRFs. It has been well studied that aglycones are more active forms in intestines, resulting in higher bioavailability than their glycoside forms [31,32]. Several studies have also revealed that deglycosylated phytochemicals enhance biological activities. For example, ginseng saponin deglycosylated by heat processing shows improved anticancer activities [33,34]. The aglycone forms in dietary soy isoflavones displayed beneficial effects on lipid metabolism in rats [35]. On the other hand, most of the natural polyphenols are found in plants as free or bound forms [36]. The inclusion of an acid, rather than alkaline, hydrolysis step results in higher antioxidant capacity and extraction efficiency of bound phenolic compounds by releasing free phenolics [37]. Moreover, our previous study showed that the conversion of glycosidic flavonoids to aglycones by acid treatment increases its bioactivity [14]. Therefore, acidic hydrolysis could be effective for increasing the bioactivity of phenolic compounds by deglycosylation and/or releasing bound phenolic compounds. These results showed that deglycosylation of rutin by acid treatment afforded anti-obesity and hypolipidemic activities in obese mice in vivo.

## 4. Conclusions

In conclusion, AR treatment decreased body and epididymal fat weight in HFD group mice by the conversion of rutin to quercetin. AR also ameliorated lipid metabolic parameters such as levels of TC, TG, LDL-c, and HDL-c in serum samples. These results showed that acid treatment of rutin enhanced anti-obesity properties in in vivo systems. Taken together, these results demonstrated that deglycosylation of flavonoids contributes to improved bioactivity, including anti-obesity and hypolipidemic properties. Acid hydrolysis is a relatively simple process and also can be applied to various plant materials that contain abundant flavonoid glycosides. These results provide a new insight into enhancing the functional properties of flavonoids in foodstuffs upon acid hydrolysis. Finally, further research is ongoing to determine the anti-obesity and antioxidant effects of deglycosylation in Tartary buckwheat, which is a rich source of rutin.

## Figures and Tables

**Figure 1 nutrients-12-01539-f001:**
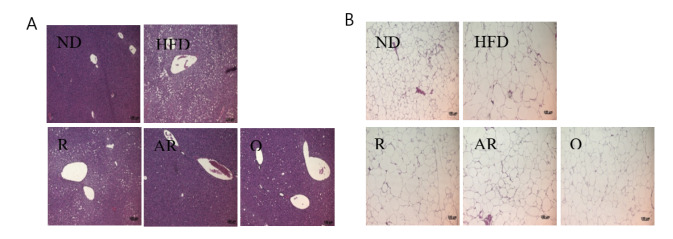
Histological analysis of liver (**A**) and epididymal (**B**) tissue in mice fed with experimental diets for 7 weeks. ND, normal diet; HFD, high-fat diet; R, high-fat diet with 100 mg/kg rutin/day; AR, high-fat diet with 100 mg/kg acid-treated rutin/day; Q, high-fat diet with 100 mg/kg quercetin/day.

**Figure 2 nutrients-12-01539-f002:**
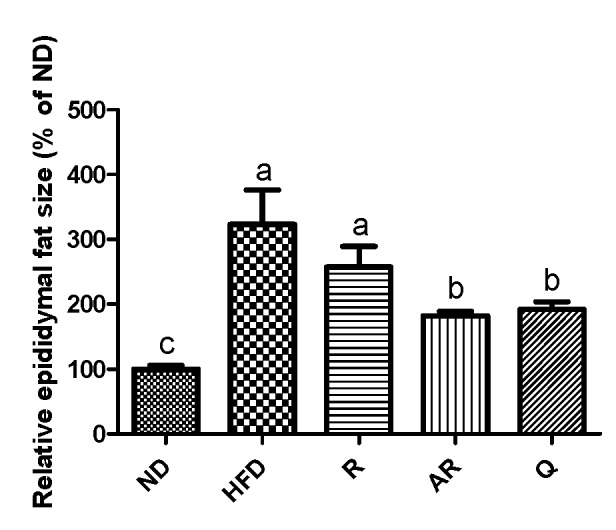
Effect of experimental diets on relative epididymal adipocyte size in mice fed with experimental diets for 7 weeks. The results are expressed as means ± standard deviation. ^a-c^ Different letters above bars mean significant differences (*p* < 0.05). ND, normal diet; HFD, high-fat diet; R, high-fat diet with 100 mg/kg rutin/day; AR, high-fat diet with 100 mg/kg acid-treated rutin/day; Q, high-fat diet with 100 mg/kg quercetin/day.

**Figure 3 nutrients-12-01539-f003:**
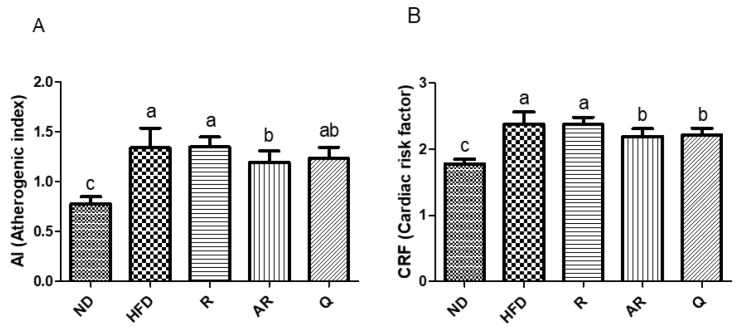
Effect of experimental diets on atherogenic index (**A**) and cardiac risk factor (CRF) (**B**) in mice fed with experimental diets for 7 weeks. Atherogenic index (AI) = (total cholesterol - high density lipoprotein cholesterol)/ high density lipoprotein cholesterol; CRF = total cholesterol/high density lipoprotein cholesterol. The results are expressed as means ± standard deviation. ^a-c^ Different letters above bars represent significant differences (*p* < 0.05). ND, normal diet; HFD, high-fat diet; R, high-fat diet with 100 mg/kg rutin/day; AR, high-fat diet with 100 mg/kg acid-treated rutin/day; Q, high-fat diet with 100 mg/kg quercetin/day.

**Table 1 nutrients-12-01539-t001:** Composition of experimental diets (g/kg diet).

Ingredient	Experimental Groups ^(1)^	
	ND	HFD
Casein	200	200
L-Cysteine	3	3
Corn starch	402.2	214.7
Maltodextrin 10	70	100
Sucrose	172.8	172.8
Cellulose	50	50
Soybean oil	25	25
Lard	20	177.5
Mineral Mix S10026	10	10
Dicalcium phosphate	13	13
Calcium carbonate	5.5	5.5
Potassium citrate	16.5	16.5
Vitamin Mix V10001	10	10
Choline Bitartrate	2	2

^(1)^ The diets were prepared to be isocaloric (473 kcal/100 g) in experimental groups except normal diet (385 kcal/100 g) (originally formulated by E.A. Ulman, Ph.D., Research Diets, Inc., Jules Lane New Brunswick, NJ, USA). ND, normal diet; HFD, high-fat diet.

**Table 2 nutrients-12-01539-t002:** Effect of experimental diets on weekly changes on body weight in mice.

Groups	ND (g/mice)	HFD (g/mice)	R (g/mice)	AR(g/mice)	Q (g/mice)
0 day	22.95 ± 1.45 ^Ba^	23.60 ± 1.30 ^Ba^	22.72 ± 0.73 ^Ba^	22.80 ± 1.39 ^Ba^	23.13 ± 0.94 ^Ba^
3 days	23.40 ± 1.55 ^Ba^	26.52 ± 1.48 ^Ba^	25.14 ± 1.26 ^Ba^	24.37 ± 1.68 ^Ba^	24.64 ± 1.49 ^Ba^
7 days	24.14 ± 1.71 ^Ba^	27.43 ± 1.33 ^Ba^	26.53 ± 1.51 ^Ba^	25.41 ± 1.65 ^Ba^	25.66 ± 1.54 ^Ba^
14 days	24.55 ± 1.74 ^Ba^	28.39 ± 1.27 ^Aa^	27.40 ± 1.37 ^Aa^	26.52 ± 2.12 ^Aa^	26.16 ± 1.86 ^Ba^
21 days	24.90 ± 1.91 ^Bb^	30.10 ± 1.56 ^Aa^	28.77 ± 1.47 ^Aab^	27.40 ± 2.14 ^Aab^	27.66 ± 2.02 ^Aa^
28 days	24.76 ± 2.16 ^Bb^	32.00 ± 2.19 ^Aa^	29.62 ± 1.97 ^Aa^	28.83 ± 2.75 ^Aa^	28.54 ± 1.98 ^Aab^
35 days	26.28 ± 2.17 ^Bb^	33.70 ± 2.71 ^Aa^	30.99 ± 2.16 ^Aa^	29.67 ± 3.27 ^Aab^	29.83 ± 1.62 ^Aab^
42 days	27.70 ± 1.79 ^Ab^	35.17 ± 2.70 ^Aa^	32.28 ± 2.94 ^Aa^	31.04 ± 3.43 ^Ab^	31.15 ± 2.02 ^Aa^
49 days	27.33 ± 3.47 ^Ac^	38.12 ± 3.29 ^Aa^	34.07 ± 3.08 ^Ab^	32.73 ± 4.10 ^Ab^	33.01 ± 2.42 ^Ab^

ND, normal diet; HFD, high-fat diet; R, high-fat diet with 100 mg/kg rutin/day; AR, high-fat diet with 100 mg/kg acid-treated rutin/day; Q, high-fat diet with 100 mg/kg quercetin/day. The results are expressed as means ± standard deviation. ^A-B^ Means in the same column followed by different capital letters differ significantly. ^a-c^ Means in the same row followed by different small letters differ significantly.

**Table 3 nutrients-12-01539-t003:** Effect of experimental diets on body weight gain, food intake, and food efficiency ratio (FER) in mice.

Group	Body Weight Gain (g/Mice/7 Weeks)	Food Intake (g/Mice/Day)	FER (%) ^(1)^
ND	4.37 ± 2.83 ^c^	3.09 ± 0.41 ^a^	2.51 ± 3.06 ^b^
HFD	14.52 ± 3.42 ^a^	2.70 ± 0.13 ^b^	8.96 ± 3.89 ^a^
R	11.36 ± 3.35 ^b^	2.55 ± 0.18 ^b^	7.23 ± 1.94 ^a^
AR	9.92 ± 4.48 ^b^	2.47 ± 0.24 ^b^	7.31 ± 2.32 ^a^
Q	9.87 ± 2.49 ^b^	2.52 ± 0.22 ^b^	7.04 ± 2.90 ^a^

^(1)^ FER (%) = (body weight gain (g/day)/food intake (g/day)) × 100. ND, normal diet; HFD, high-fat diet; R, high-fat diet with 100 mg/kg rutin/day; AR, high-fat diet with 100 mg/kg acid-treated rutin/day; Q, high-fat diet with 100 mg/kg quercetin/day. The results are expressed as means ± standard deviation for seven mice in each group. ^a-c^ Values with different superscript letters within the same column differ significantly (*p* < 0.05).

**Table 4 nutrients-12-01539-t004:** Effect of experimental diets on liver and epididymal fat weight in mice.

Group	Liver (g)	Epididymal Fat (g)
ND	1.33 ± 0.15 ^b^	0.65 ± 0.24 ^c^
HFD	1.56 ± 0.15 ^a^	2.56 ± 0.40 ^a^
R	1.20 ± 0.13 ^bc^	2.08 ± 0.49 ^b^
AR	1.14 ± 0.17 ^c^	1.94 ± 0.44 ^b^
Q	1.21 ± 0.11 ^bc^	2.05 ± 0.42 ^b^

ND, normal diet; HFD, high-fat diet; R, high-fat diet with 100 mg/kg rutin/day; AR, high-fat diet with 100 mg/kg acid-treated rutin/day; Q, high-fat diet with 100 mg/kg quercetin/day. The results are expressed as means ± standard deviation for seven mice in each group. ^a-c^ Values with different superscript letters within the same column differ significantly (*p* < 0.05).

**Table 5 nutrients-12-01539-t005:** Effect of experimental diets on plasma biochemical parameters in mice.

Parameters	ND	HFD	R	AR	Q
AST (IU/L)	94.14 ± 22.79 ^b^	133.81 ± 13.51 ^a^	94.59 ± 49.26 ^b^	126.26 ± 39.40 ^ab^	97.43 ± 47.31 ^b^
ALT (IU/L)	29.51 ± 4.58 ^c^	59.16 ± 20.44 ^a^	38.31 ± 10.17 ^c^	30.87 ± 9.78 ^bc^	41.53 ± 15.65 ^b^
BUN (mg/dL)	26.08 ± 4.75 ^a^	24.10 ± 5.73 ^a^	25.97 ± 5.39 ^a^	23.19 ± 3.72 ^a^	21.75 ± 3.72 ^a^
CREA (mg/dL)	0.44 ± 0.03 ^a^	0.43 ± 0.01 ^a^	0.41 ± 0.02 ^b^	0.40 ± 0.02 ^b^	0.40 ± 0.02 ^b^
GLC (mg/dL)	233.14 ± 31.63 ^b^	270.46 ± 38.89 ^ab^	274.04 ± 54.32 ^a^	272.72 ± 34.49 ^ab^	255.39 ± 37.89 ^ab^
TG (mg/dL)	45.49 ± 11.05 ^bc^	69.30 ± 20.33 ^a^	54.10 ± 13.05 ^ab^	36.82 ± 12.10 ^cd^	28.23 ± 11.26 ^d^
TC (mg/dL)	119.95 ± 11.86 ^c^	253.43 ± 24.68 ^a^	263.00 ± 22.13 ^a^	222.89 ± 26.42 ^b^	243.92 ± 23.85 ^a^
LDL-c (mg/dL)	10.90 ± 1.85 ^d^	27.26 ± 3.90 ^a^	26.26 ± 3.59 ^a^	18.58 ± 3.61 ^c^	22.60 ± 3.92 ^b^
HDL-c (mg/dL)	67.66 ± 6.56 ^c^	110.16 ± 9.86 ^ab^	111.86 ± 8.24 ^a^	102.63 ± 7.61 ^b^	106.23 ± 11.58 ^ab^

ND, normal diet; HFD, high-fat diet; R, high-fat diet with 100 mg/kg rutin/day; AR, high-fat diet with 100 mg/kg acid-treated rutin/day; Q, high-fat diet with 100 mg/kg quercetin/day. AST, aspartate aminotransferase; ALT, alanine aminotransferase; BUN, blood urea nitrogen; CREA, creatinine; GLC, glucose; TG, triglycerides; TC, cholesterol; LDL-c, low density lipoprotein cholesterol; HDL-c, high density lipoprotein cholesterol. The results are expressed as means ± standard deviation. ^a-d^ Values with different superscript letters within the same row differ significantly (*p* < 0.05).

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
