# Peer review of "Effect of Deglycosylated Rutin by Acid Hydrolysis on Obesity and Hyperlipidemia in High-Fat Diet-Induced Obese Mice"

_nutrients, 2020, doi:10.3390/nu12051539_

Round 1

Reviewer 1 Report

Generally, the biochemical and histological experiment are well done but the novelty is an issue. There is a wide selection of relevant literature on quercetin antiobesity effects in animals or humans. Previously conducted animal studies showed that quercetin, a product of rutin hydrolysis, protect both mice and rats from HFD induced BWG and adipose tissue accumulation. It is well known that in HFD fed animal model quercetin suppressed adipogenesis and lipogenesis. Mechanism behind quercetine role in obesity treatment is known (Role of Quercetin as an Alternative for Obesity Treatment: You Are What You Eat! Food Chem 2015, 179:305-10).

In Introduction wide range of literature on quercetin antiobesity effect should be summarized shortly. Respectable studies focused on antiobesity effect of quercetin should be mentioned and for readers’ convenience mechanism behind could be shortly explained. Differences between approach conducted in submitted study and previously published trials related to quercetin and/or rutin effects in treatment of obesity needs to be emphasized with the aim to clearly explain the reason to conduct experiments and also to indicate novelty expecting to emerge.

In Discussion differences between results of previously published trials related to quercetin effects and presented results should be stressed and novel insight into the quercetin role in obesity treatment obtained clearly stated. Novelty needs to be explained in light of obtained results comparison with previous studies on quercetin. Comparison of obtained data with literature ones is essential to define novelty. If the reference study was in rats rather than mice, the reference values must be adjusted using an appropriate dose translation factor (https://doi.org/10.1096/fj.07-9574LSF or DOI: 10.1002/ddr.21461). In Conclusion novelty needs to be clearly defined.

There are several issues related to description of experimental details and results presentation.

MM

At the beginning of MM there is an omission to provide common information about chemicals used. Please provide usual information about all chemicals used under the subtitle Chemicals including producers as well as purity of rutin and quercetin used. Components of diet need to be included as well. Please provide information related to quantification of Q in AR and its purity. It is not enough to cite previous paper.  More detailed data on diet preparation and administration could be given. Please describe exactly how the diet was prepared. Please explain if AR dissolved in DMSO was mixed with diet components and then the whole diet dissolved in CM cellulose and if the same procedure was applied to introduce Q and R? Why intake of water was not measured. Water intake should be given as well. Why the period of 7 weeks was considered? Are there any records of illness or death among the mice? How was the dosage chosen? If based on previous studies please cite it.

RD

In most of the results there is no significant difference between the R and AR groups to approve conclusion that "acid-treated routine" had the more prominent effect than the routine itself?

The description of the statistics is inadequate. It should be conducted correctly according to: https://elifesciences.org/articles/36163. Also, changes need to be introduced into figures and tables according to corrected statistics.

Row 118-119 The sentence is not clear, it is rather confusing.

Table 2. Units are missing. Two-way repeated ANOVA should be applied to this table. The name of the table is inappropriate. It is not only the effect of "acid-treated routine" but comparisons between effects of diets (the whole diet). The same comment is applicable to other tables and figures entitled in the same way.

Table 3. The increase in mass is not equal to the difference between the initial and final mass of each group. Please explain this discrepancy. Also, if food consumption was measured once a week than daily ANOVA data does not exist and weekly data can not be used for this purpose.

Table 5. There is a value for HDL 11, 86 for group R, while for other groups the values ​​are 60-100. Please include some comment or correct the mistake. Also, check the numbers in the whole table and correct the spelling mistakes.

The influence of food matrix on quercetin stability and bioavailability should be considered in future studies. Papers entitled Bioavailability of quercetin in humans and the influence of food matrix comparing quercetin capsules and different apple sources in Foods Reasearch International, 88A, 2016, 159 and The biological activities, chemical stability, metabolism and delivery systems of quercetin: A review, published in Trends in Food Science and Technology, 56, 2016, 21 can be considered.

Author Response

Thank you very much for your consideration, and we really appreciate the comments and have learned a lot. We have revised the manuscript according to your suggestions. The changes were marked with red color in the revised manuscript. Please chesk the attachment.

Thank you so much.

Reviewer 2 Report

This is a relatively small mouse study of the potential anti-obesity impact of acid-treated rutin. English grammar requires some minor attention throughout. Mouse numbers per treatment group are minimal (n=7) although variability is surprisingly low. Some broader discussion of related plant-sourced extracts (eg Grape Seed Extract) would be useful.

ABSTRACT:

'smaller by about' is non-scientific

p-values or similar should be included to illustrate magnitude of changes

line 20: 'providing valuable information'

INTRODUCTION:

A good summary of the area and study aims.

line 46: 'to compare the potential anti-obesity....'

METHODS;

Statistical justification for n=7 mice should be provided.

line 51: Rutin (5oo mg) was dissolved....

line 69: di=2. It is not possible to 'divide into 5'

line 103: Why are both SD and SEM presented??

RESULTS and DISCUSSION:

Why are both SD and SEM presented??

Include some discussion on other plant-sourced compounds (eg procyanidins).

Author Response

Thank you very much for your work on my manuscript. I have revised the manuscript according to your suggestions. The changes were marked in the revised manuscript.

Thank you so much.

Reviewer 3 Report

The article focuses on comparison of supplementation with selected flavonoids (rutin, acid-treaded rutin and quercetin) on anti-obesity and hypolipidemic effects in obese mice. Submitted manuscript is continuation of previously published data regarding acid treated rutin in in vitro study. The introduction includes necessary information and properly leads to the aim of the study. Material and methods are sufficiently detailed for proper understanding of the experiment. Discussion properly facilitates the reader with understanding of provided experimental data and is supported by the results. The authors also did not forget to refer to the literature throughout the discussion. Provided literature is sufficient covering past 20 years in the field of research. Overall submitted manuscript is a valuable addition to the field of research regarding the effect of antioxidants in obesity.

Submitted manuscript  could be almost published in present form, however I have found some minor issues that should be checked prior the publication:

  • There is a lack of auxiliary verb between line 45 and Line 46,
  • Please reconsider placing a X Y graph instead of table 2 to facilitate the reader , in current form its almost unreadable
  • Table 5 – last row, data in column 3 i 4 is wrong

Author Response

Thank you very much for your work on my manuscript. I have revised the manuscript according to your suggestions. The changes were marked with red color in the revised manuscript.

Thank you so much.

Round 2

Reviewer 1 Report

Some mistakes are corrected and partial improvements are introduced but major issues are not resolved. Experiments were not compared with previously published, as requested, and evidence that paper provides novelty, required for publication, is not provided. Also, it is confirmed that statistical significance between groups, necessary to consider results relevant, does not exist. Moreover, differences between results groups are not consistent at all while difference between AR and Q requires an explanation. 

Author Response

We sincerely appreciate your comments on my work. We are sure your comments have led to a better manuscript. If our revision does not match your comments, please let us know and I will revise it again.

Please kindly find the attached file. Thank you very much.
